# Vein–Membrane Interaction in Cambering of Flapping Insect Wings

**DOI:** 10.3390/biomimetics8080571

**Published:** 2023-11-27

**Authors:** Daisuke Ishihara, Minato Onishi, Kaede Sugikawa

**Affiliations:** Department of Intelligent and Control Systems, Kyushu Institute of Technology, 680-4 Kawazu, Iizuka 820-8502, Fukuoka, Japan; oonishi.minato253@mail.kyutech.jp (M.O.); sugikawa.kaede544@mail.kyutech.jp (K.S.)

**Keywords:** flapping insect wing, camber, vein–membrane interaction, shape simplification model, monolithic solution procedure, finite element method

## Abstract

It is still unclear how elastic deformation of flapping insect wings caused by the aerodynamic pressure results in their significant cambering. In this study, we present that a vein–membrane interaction (VMI) can clarify this mechanical process. In order to investigate the VMI, we propose a numerical method that consists of (a) a shape simplification model wing that consists of a few beams and a rectangular shell structure as the structural essence of flapping insect wings for the VMI, and (b) a monolithic solution procedure for strongly coupled beam and shell structures with large deformation and large rotation to analyze the shape simplification model wing. We incorporate data from actual insects into the proposed numerical method for the VMI. In the numerical analysis, we demonstrate that the model wing can generate a camber equivalent to that of the actual insects. Hence, the VMI will be a mechanical basis of the cambering of flapping insect wings. Furthermore, we present the mechanical roles of the veins in cambering. The intermediate veins increase the out-of-plane deflection of the wing membrane due to the aerodynamic pressure in the central area of the wing, while they decrease it in the vicinity of the trailing edge. As a result, these veins create the significant camber. The torsional flexibility of the leading-edge veins increases the magnitude of cambering.

## 1. Introduction

Insects were the first organisms on Earth to fly. Because flight is superior to other forms of locomotion, flying insects have extended their habitat to all over the world [1]. Their flight abilities have become increasingly sophisticated through the evolutionary process [2]. Therefore, the development of micro and nano aerial vehicles that mimic the flapping flight of insects has attracted the attention of many researchers.

Insects’ flapping flight consists of multiple systems, such as neural networks, sensory mechanisms, musculoskeletal morphology and actuation, and wing kinematics and aerodynamics, which span a wide range of time and length scales [3,4,5]. These systems are strongly coupled to create sophisticated flight behavior.

This study focuses on mechanical systems. Many researchers have attempted to understand the wing kinematics and aerodynamic mechanisms that enable insects to generate the forces and torques required to fly and maneuver. The aerodynamic basis of this understanding is the leading-edge vortex created by the flapping of an insect’s wing [6,7,8] with the characteristic feathering motion and the tip path, such as a figure eight [9,10,11].

The articulation at the base of the insect wing forms a complicated structure, consisting of sclerites and muscles. This articulation works as the transmission, which redirects the driving force and torque from the main flight muscles to the wing. Hence, this articulation enables both active and passive controls of characteristic wing’s motions. The wing’s flapping motion is regulated at this articulation [10,12,13], while the feathering motion and wingtip path can be generated based on the interaction between the flapping flexible wing and the surrounding air [10,14,15,16,17].

Furthermore, insect wings undergo cambering in several orders of insects during flapping [2,9,14,18,19]. The cambering is considered to result from elastic deformation of the wings due to aerodynamic forces [14] because the wings that consist of veins and membranes do not have any interior muscle [19,20]. The camber increases the aerodynamic performance of the insect’s flapping flight [21,22,23,24]. Therefore, clarifying the mechanical process of cambering or how elastic deformation of flapping insect wings caused by the aerodynamic pressure results in their significant camber is important. The mechanical process of cambering will provide design guidance for insect-inspired micro and nano aerial vehicles and passive shape control of lightweight structures subject to aerodynamic forces. However, it seems that the mechanical process of cambering is still unclear.

Hence, in this study, we present a vein–membrane interaction (VMI) to clarify this mechanical process. In order to investigate the VMI, we propose a numerical method that consists of (a) a shape simplification model wing that consists of a few beams and a rectangular shell structure as the structural essence of flapping insect wings for the VMI, and (b) a monolithic solution procedure for strongly coupled beam and shell structures with large deformation and large rotation to analyze the shape simplification model wing.

Various models of insect wings have been developed, from conceptual models to realistic models [19]. Realistic models are intended to replicate structural details using finite elements [25,26,27,28,29]. However, insect wings vary widely in their structures [30], ranging from macroscopic complexity, such as networks of veins and corrugation structures [31,32,33], to microscopic complexity, such as various types of sectional shapes of veins [34], sandwich structures of veins [35], additional strengthening of small veins to significant veins [34], spikes [36], vein-joints [37], and the nodus [38]. The influences of these structures on the wing’s macroscopic behaviors such as the cambering have been investigated using realistic models from the mechanical viewpoint. This approach might lead to a model wing that can directly reproduce a camber equivalent to that of the actual insect. However, it seems that there is no such model wing at this time. This is mainly because of computational difficulties in the methodological level as well as the computational cost to replicate all necessary structural details using finite elements.

The conceptual analogy for insect wings is an oscillating sail, which has the sailcloth, the mast and boom, and the battens, which correspond to, respectively, the wing membrane, the two major supporting domains of the wing, and the stiffening veins [19]. In geometrical model wings, the straight cylinder representing the leading edge, which is twisted linearly about the longitudinal axis, can rotate a series of lines representing veins, which diverge from the surface of the cylinder, such that the face consisting of them forms the camber [14]. In these models, the explicit usage of the veins is key for explaining the cambering.

The wing membrane catches the aerodynamic pressure, and it transmits this force to the veins. Then, the veins are deformed, and at the same time they support the wing membrane. Finally, the wing forms specific shapes such as the camber as a result of the interior balance between the veins and the membrane under the aerodynamic force. In this study, we consider this mechanical process as the structure–structure interaction [39], specifically the vein–membrane interaction (VMI). As mentioned above, the conceptual models have explained cambering by incorporating explicit structures corresponding to the veins. However, they have difficulty in describing the mechanical process of the VMI. As other scenarios, the wing flexion lines due to wing folding in locusts [40], the resilin patch/stripe distribution in damselflies and bees [41,42], and the forewing–hindwing coupling in bees [43] have been proposed, which could significantly contribute to the wing cambering. However, the wings appeared to camber in a gentle curve, not localized to flexion lines [44].

In previous structural models [27,28,45], the in-plane variation in mechanical properties was given to the finite elements for the purpose of modeling anisotropic stiffness variations in insect wings. However, as suggested by the conceptual models, explicit structures corresponding to veins and wing’s membranes will be necessary for the structural model wing to simulate significant cambers. In this study, as an essence for this purpose, a shape simplification model wing that consists of a few beams and a rectangular shell, which are tightly attached, is proposed.

In the shape simplification model wing, the supporting functions of the veins branching radially from the base to the tip are roughly classified into two major supports in the leading-edge and trailing-edge areas and the intermediate support for the wing’s interior area between these areas. These supports are represented by the leading-edge beam, the trailing-edge beam, and the intermediate beam, respectively, and they are connected to each other at the base. The wing’s membrane supported by the veins is represented by a rectangular shell structure. The beams and the shell are tightly attached to each other. Hence, different from the previous models, the proposed model can describe the VMI using this coupling of the beam and shell structures. The present modeling belongs to a reduced-order approach, which gives a framework that characterizes the complicated behaviors and a well-defined functional target and can guide in determining a mechanism for the behavior [13].

For the purpose of simplicity, this study focuses on the middle of each half-stroke since the cambering is most significant at this point. In this case, the quasi-steady approximation can be used adequately since the equilibrium between the wing’s elastic force and the dynamic pressure is dominant at this point [17]. The shape simplification model wing is discretized using compatible beam and shell finite elements, and their equilibrium equations are formulated by the nonlinear finite element method that can accurately analyze the nonlinear relationship between the model wing’s large deformation and large rotation and the applied forces [46]. The strongly coupled beam and shell structures are formulated using the monolithic method [47,48].

We incorporate data from actual insects [49] into the proposed numerical method that consists of the shape simplification model wing and the nonlinear monolithic solution procedure. In the numerical analysis, we demonstrate that the model wing can generate a camber equivalent to that of the actual insects. Hence, the VMI will be a mechanical basis of the cambering of flapping insect wings. Furthermore, the mechanical roles of the veins in the cambering are presented as follows: The intermediate veins increase and decrease the out-of-plane deflection of the wing membrane due to the aerodynamic pressure in, respectively, the central area and the vicinity of the trailing edge. As a result, the intermediate veins create significant cambering. The torsional flexibility of the leading-edge veins increases the magnitude of cambering.

## 2. Modeling of Flapping Insect Wings

Insect wings consist of veins and membranes, and they do not have any interior muscle [14,19,20,50]. Hence, in flapping wings, the equilibrium between the veins and membrane under aerodynamic pressure results in cambering. We consider this mechanical process as the VMI. As suggested by the previous conceptual models [14,19], explicit structures corresponding to veins and wing’s membranes will be necessary for structural models to simulate cambering. Here, we propose a shape simplification model wing that consists of a few beams and a rectangular shell structure, which are tightly attached to each other, as the structural essence to describe the VMI.

### 2.1. Shape Simplification Model Using Beam and Shell Structures

We considered the dipteran wing since it has been used frequently in previous studies on flight mechanisms. There exist two major supporting areas in the leading edge and the trailing edge [19,50]. Each area consists of several significant veins that work together to support the wing. The behavior of these areas can be characterized by their macroscopic bending and torsion since their bodies are slender. Hence, their supports are represented by beam structures, that is, the leading-edge beam for the leading-edge support and the trailing-edge beam for the trailing-edge support, of which the stiffnesses are determined by their macroscopic constitutive relations. In the interior area between two major supporting areas, the veins branch radially from the wing’s base to the trailing edge to support the wing’s membrane. This intermediate support is represented by a beam structure (intermediate beam). In addition to these major supports (leading edge, trailing edge, and intermediate supports), a supplementary support given by veins in the areas among them is represented by the supplementary beam. The wing’s membrane is considered as a shell structure. All beam structures are tightly attached to the shell structure.

The wing’s deformation is considered at the middle of each half-stroke since the cambering is most significant at this point. In this case, the quasi-steady approximation can be used adequately since the equilibrium between the wing’s elastic force and the dynamic pressure are dominant in the proximity of this point [17], where the flapping velocity is approximately constant. The quasi-steady model of the aerodynamic forces is described in the following section. The total pressure is proportional to the square of the nondimensional radius of the second moment of the wing area *r*_2_ [51]. The *r*_2_ value for a rectangular wing is very close to that of an actual insect’s wings. Therefore, the rectangular wing with the aspect ratio *R*_A_ of the actual insect’s wing is used for the purpose of simplicity, where *R*_A_ is defined as 2*R*_w_/*C*_w_^m^ (*R*_w_: one wing span length, *C*_w_^m^: mean chord length).

It follows from these simplifications that a shape simplification model is obtained, as shown in Figure 1, where beam finite elements are used for the discretization of beam structures and shell finite elements are used for the discretization of the shell structure. The finite element formulation for strongly coupled beam and shell structures with large displacements and large rotations is described in the following section. Note that the purpose of the proposed model was simulating macroscopic mechanical behaviors of flapping insect wings. Hence, microscopic mechanical properties such as the local stiffness might not be accurately formulated. If the local stiffness is not accurately formulated, higher mode vibrations in flapping wings will not be accurately simulated. However, the higher mode vibrations, which are induced by the stroke reversal at the start of each half-stroke, are quickly damped in the half-stroke [17].

### 2.2. Quasi-Steady Model of Aerodynamic Forces

A flapping wing model is shown schematically in Figure 2, where a hovering flight is considered as the flight regime. As shown, the flapping axis coincides with the *y*-axis, and the leading edge flaps and translates in the *xz*-plane to draw a stroke plane with the stroke angle *Φ*. The flapping direction in this figure is assumed to be clockwise about the *y*-axis. The flapping angular displacement *φ* is defined as the angle between the leading edge and the *x*-axis, and it varies in the range between −*Φ*/2 and *Φ*/2. The camber is considered at the middle of each half-stroke or *φ* = 0 because it appears most significantly near the middle of each half-stroke in actual insects [49].

The recoil of cambering can be observed in actual insects during reverse strokes, but this inertial effect quickly disappears [49] because of significant air damping [17]. Furthermore, among the aerodynamic forces acting on the insect’s wings, the aerodynamic pressure is most dominant [8]. Therefore, only camber caused by the aerodynamic pressure is considered.

Different from the linear translation of a two-dimensional wing forming a von Karman vortex street, the flapping translation of a three-dimensional wing forms a stable leading-edge vortex due to the existence of axial flow [52], which generates stable aerodynamic pressure acting on the wing [8]. The flapping speed is approximately constant in the middle of each half-stroke. Therefore, the aerodynamic pressure acting on the three-dimensional wing, which flaps with a constant speed, can be evaluated quasi-statically using an adequate fluid force coefficient [53].

The time history of the flapping angular velocity can be approximated using a trapezoidal function, as shown in Figure 3 [54], which is based on actual observations [9,55]. The speed of the flapping translation is constant and has a maximum magnitude *ω*_max_ in the middle of each half-stroke (*φ* = 0), as shown in this figure. Therefore, in this study, the aerodynamic pressure was evaluated using the following quasi-static equations:(1)$$P=\frac{1}{2}C_{D}\rho^{f}V_{\max}{}^{2},\;V_{\max}=\omega_{\max}r=\frac{8\Phi r}{3T_{\varphi }},$$
where *C*_D_ is the drag force coefficient for the flat plate, *ρ*^f^ is the fluid mass density, *V*_max_ is the maximum flapping speed at each point on the wing surface, *r* is the distance from the flapping axis or the *y*-axis along the wing length, *Φ* is the stroke angle, and *T_φ_* is the flapping period or the inverse of the flapping frequency *f_φ_*. The acceleration and deceleration time *t*_a_ is set as *T_φ_*/8, which is a typical value [54].

As shown in Figure 4a, the magnitude of the camber is defined as the ratio of the height between the wing chord center and wing surface to the wing chord length [49], and the camber is positive if it is concave along the direction of the flapping translation. As shown in Figure 4b, the magnitude of the twisting is defined as the slope angle of the current wing chord to the initial wing chord (feathering angle *θ*), and anti-clockwise twisting along the length is positive.

## 3. Monolithic Solution Procedure for Strongly Coupled Beam and Shell Structures with Large Deformations and Large Rotations

### 3.1. Governing Equations for the Body with Large Displacements and Large Rotations

A fundamental difficulty in the general application of the equilibrium of the body with large deformations and large rotation is that the configuration of the body is unknown or is changing continuously. This change can be dealt with by defining auxiliary stress and strain measures. The principle of virtual displacements in the total Lagrangian formulation is given by
(2)∫V0S0t+Δtijδε0t+ΔtijdV0=Rt+Δt,
where _0_*^t^S_ij_* is the *ij*-th component of the second Piola–Kirchhoff stress tensor at time *t*, *δ*_0_*^t^ε_ij_* is the *ij*-th component of the Green–Lagrange strain tensor at time *t* corresponding to the virtual displacement, ^0^*V* expresses the initial volume of the body, and *^t^R* is the external virtual work at time *t*. The left-hand side is the internal virtual work. In a static analysis without time effects other than the definition of the load level, time is only a convenient variable which denotes different intensities of load applications and correspondingly different configurations. The important aspect of this equation is that the integration is performed over the initial volume of the body to avoid the above difficulty.

Let us fully linearize this expression with respect to a general finite element nodal degree of freedom *^t^a_k_*, which can be a displacement or rotation, since the principle of virtual displacements (2) is for elements with rotational degrees of freedom, as shown in the next section. Using a Taylor series expansion,
(3)Sij0t+Δtδεij0t+Δt≐Sij0tδεij0t+∂∂aktSij0tδε0tijdak,
where *da_k_* is a differential increment in *^t^a_k_*. The substitution of Equation (3) into the principle of virtual displacements (2) gives
(4)∫V0∂∂atkS0tijδε0tijdV0dak=Rt+Δt−∫V0S0tijδε0tijdV0.

Note that we use
(5)δε0tij=∂ε0tij∂atlδal,
where *δa_l_* is a variation in *^t^a_l_*, and hence the variation is taken with respect to the nodal parameter *^t^a_l_* about the configuration at time *t*, and the term appearing in the left-hand side of Equation (4) can be chain-differentiated to obtain
(6)∂∂atkS0tijδε0tij=∂S0tij∂ε0trs∂ε0trs∂atk∂ε0tij∂atlδal+S0tij∂∂atk∂ε0tij∂atlδal=Cijrs0∂ε0trs∂atk∂ε0tij∂atlδal+S0tij∂2ε0tij∂atk∂atlδal,
where in the last term
(7)∂S0tij∂εrs0t=Cijrs0,
is used. A simple and widely used elastic material description for large deformation analysis is obtained by generalizing the linear elastic relations to the total Lagrangian formulation as _0_*^t^S_ij_* = _0_*C_ijrs_* _0_*^t^ε_rs_*, where _0_*C_ijrs_* is the *ijrs*-th component of the constant elasticity tensor.

Substitution of Equations (5) and (6) into the linearized principle of virtual displacements (4) gives
(8)∫V0Cijrs0∂ε0trs∂atk∂ε0tij∂atldV0+∫V0S0tij∂2ε0tij∂atk∂atldV0dakδal=Rt+Δtl−∫V0S0tij∂ε0tij∂atldV0δal,
where *^t^*^+Δ*t*^*R_l_* denotes the external virtual work corresponding to *δa_l_*.

### 3.2. Compatible Beam and Shell Finite Elements

The beam structures that represent the supports from the veins are discretized using the beam elements, while the shell structure that represents the wing’s membrane is discretized using the shell elements, and they are strongly coupled or satisfy the coupling condition in their interface. The coupling condition consists of compatibility and equilibrium conditions. For monotonic convergence, the finite elements must be complete and the finite elements and mesh must be compatible [46]. This means mathematically that the new space of finite element interpolation functions will contain the previously used space, and as the mesh is refined, the dimension of the finite element solution space will be continuously increased to ultimately contain the exact solution.

The requirement of completeness of an element means that the displacement functions of the element must be able to represent the rigid body displacements and the constant strain states. This requirement is naturally satisfied in commonly used finite elements. The requirement of compatibility means that the displacements within the elements and across the element boundaries must be continuous. Compatibility ensures that no gaps occur between elements when the assemblage is loaded. Hence, we impose the requirements on beam and shell elements, as shown in Figure 5. A shell element with six degrees of freedoms (DOFs) at each node is selected following nodal DOFs of a general curved beam element. The node arrangements of beam and shell elements are the same, and these elements use the same polynomial orders of interpolation on their interface.

### 3.3. Nonlinear Analysis Framework

The basic problem in a general nonlinear analysis is to find the state of equilibrium of a body corresponding to the applied loads. Assuming that the externally applied loads are described as a function of time, the equilibrium conditions of a system of finite elements representing the body under consideration can be expressed as
(9)Rt+Δt−Ft+Δt=0,
where the vector *^t^***R** lists the externally applied nodal point forces in the configuration at time *t* and the vector *^t^***F** lists the nodal point forces that correspond to the element stresses in this configuration. The basic approach in an incremental step-by-step solution is to assume that the solution for the discrete time t is known and that the solution for the discrete time *t* + Δ*t* is required, where Δ*t* is a suitably chosen time increment. Since the solution is known at time *t*, we can write
(10)Ft+Δt=Ft+ΔF,
where Δ**F** is the increment in nodal point forces corresponding to the increment in element displacements and stresses from time t to time *t* + Δ*t*. This vector can be approximated using a tangent stiffness matrix *^t^***K** which corresponds to the geometric and material conditions at time *t*,
(11)ΔF≐KtΔU,
where **U** is a vector of incremental nodal point displacements and
(12)Kt=∂Ft∂Ut.

Hence, the tangent stiffness matrix corresponds to the derivative of the internal element nodal point forces *^t^***F** with respect to the nodal point displacements *^t^***U**.

Substituting Equations (10) and (11) into Equation (9), we obtain
(13)KtΔU=Rt+Δt−Ft,
and solving for Δ**U**, we can calculate an approximation to the displacements at time *t* + Δ*t*,
(14)Ut+Δt≐Ut+ΔU.

We calculate in Equation (14) only an approximation to the exact displacements at *t* + Δ*t* that correspond to the applied loads at *t* + Δ*t* because Equation (11) was used. Hence, it is therefore necessary to iterate until the solution of Equation (9) is obtained to sufficient accuracy. Here, the iteration method is based on the Newton–Raphson technique, which is an extension of the incremental technique given in Equations (13) and (14). That is, having calculated an increment in the nodal point displacements, which defines a new total displacement vector, we can repeat the incremental solution presented above using the currently known total displacements instead of the displacements at time *t*.

The equations used in the Newton–Raphson iteration are, for *i* = 1, 2, 3, …,
(15)K(i−1)t+ΔtΔU(i)=Rt+Δt−F(i−1)t+Δt, U(i)t+Δt=U(i−1)t+Δt+ΔU(i),
with the initial conditions
(16)U(0)t+Δt=Ut, K(0)t+Δt=Kt, F(0)t+Δt=Ft.

Note that in the first iteration, the relations in Equation (15) are reduced to Equations (13) and (14). Then, in subsequent iterations, the latest estimates for the nodal point displacements are used to evaluate the corresponding element stresses and nodal point forces ^*t*+Δ*t*^**F**^(*i*−1)^ and the tangent stiffness matrix *^t^*^+Δ*t*^**K**^(*i*−1)^.

### 3.4. Monolithic Solution Procedure for Strongly Coupled Beam and Shell Structures

Substituting now the element coordinate and displacement interpolations into these equations, we obtain for assemblages of elements introduced in the previous section the equilibrium conditions of a system of finite elements representing the bodies of beam and shell elements under consideration, which correspond to Equation (9), as
(17)Rbt+Δt(Ubt+Δt)=Fbt+Δt, Rst+Δt(Ust+Δt)=Fst+Δt,
where superscripts b and s denote the quantities belonging to the beam and shell structures, respectively. These matrices and vectors depend on the specific element considered.

We impose the geometrical continuity and equilibrium conditions between the beam and shell structures. The compatible beam and shell finite elements are introduced in the previous section, where the beam elements are located along the edges of the shell elements, the beam and shell elements use the same order interpolation functions on their interface, and their node arrangements are the same. Hence, the geometrical continuity is naturally satisfied for these elements. As for the equilibrium condition, we can write
(18)$$R_{C}^{\mathrm{bs}}\equiv R_{C}^{b}+R_{C}^{s}=F_{C}^{\mathrm{bs}},$$
where the superscript bs denotes the quantity belonging to both beam and shell structures, and the subscript C denotes coupled degrees of freedom between beam and shell structures. Beam and shell displacements are decomposed into coupled and uncoupled (independent) degrees of freedom as
(19)$$U^{b}=U_{C}^{b},\;U^{s}={\left[U_{C}^{s},U_{I}^{s}\right]}^{T},$$
where the subscript I denotes uncoupled (independent) degrees of freedom, and the geometrical continuity between the beam and shell structures is described as
(20)$$U_{C}^{\mathrm{bs}}\equiv U_{C}^{b}=U_{C}^{s}.$$

Using Equations (17)–(20), the equilibrium conditions of a system of finite elements representing the body of strongly coupled beam and shell elements under consideration, which correspond to Equation (9), are
(21)RIst+Δt(UIst+Δt, UCbst+Δt)RCst+Δt(UIst+Δt, UCbst+Δt)+Rbt+Δt(UCbst+Δt)=FIst+ΔtfCbst+Δt.

Solving this equation directly is referred to as the simultaneous solution procedure [47] or the monolithic method [48]. Applying the nonlinear analysis framework, the algorithm of nonlinear analysis for strongly coupled beam and shell structures are obtained as follows:

The equations used in the Newton–Raphson iteration are, for *i* = 1, 2, 3, …,
(22)K(i−1)t+ΔtΔU(i)=Rt+Δt−F(i−1)t+Δt, U(i)t+Δt=U(i−1)t+Δt+ΔU(i),
with the initial conditions
(23)U(0)t+Δt=Ut, K(0)t+Δt=Kt,F(0)t+Δt=Ft,
where
(24)$$K=\left[\begin{array}{cc} K_{\mathrm{II}}^{s} & K_{\mathrm{IC}}^{s}\\ K_{\mathrm{CI}}^{s} & K_{\mathrm{CC}}^{s}+K_{\mathrm{CC}}^{b} \end{array}\right],\;U=\left[\begin{array}{c} U_{I}^{s}\\ U_{C}^{\mathrm{bs}} \end{array}\right],\;R=\left[\begin{array}{c} R_{I}^{s}\\ R_{C}^{\mathrm{bs}} \end{array}\right],\;F=\left[\begin{array}{c} F_{I}^{s}\\ F_{C}^{\mathrm{bs}} \end{array}\right].$$

In a static analysis without time effects other than the definition of the load level, time is only a convenient variable which denotes different intensities of load applications and correspondingly different configurations.

In the previous section, we derived the basic incremental equations used in our finite element formulations, which are the basic relations that are used in the iterations (22). Hence, we only need to focus on the basic incremental equations derived in the previous section. Substituting the element coordinate and displacement interpolations into these equations, we obtain for an assemblage of elements using the total Lagrangian formulation
(25)K0tL+K0tNLΔU=Rt+Δt−F0t,
where _0_*^t^***K**_L_ is the linear strain incremental stiffness matrix, _0_*^t^***K**_NL_ is the nonlinear strain (geometric) incremental stiffness matrix, *^t^*^+Δ*t*^**R** is the vector of externally applied nodal point loads at time *t* + Δ*t*, and _0_*^t^***F** is the vector of nodal point forces equivalent to the element stresses at time *t*. The basic integrals for the corresponding matrix evaluations are given using the formulation of the linearized principle of virtual displacements (8) as follows:(26)∫V0Cijrs0∂ε0trs∂atk∂ε0tij∂atldV0for 0tKL
(27)∫V0S0tij∂2ε0tij∂atk∂atldV0for 0tKNLand
(28)∫V0S0tij∂ε0tij∂atldV0for 0tF.

These matrices and vectors depend on the specific element considered. The displacement interpolation matrices are simply assembled from the displacement interpolation functions, and the calculation of the strain displacement and stress matrices and vectors pertaining to the continuum elements have been extensively discussed in many reports [46].

### 3.5. Basic Numerical Tests

The proposed method consists of (a) a shape simplification model wing that consists of a few beams and a rectangular shell structure as the structural essence of flapping insect wings for the VMI (see Section 2), and (b) a monolithic solution procedure for strongly coupled beam and shell structures with large deformation and large rotation (see Section 3.4) to analyze (a). In this study, it was implemented using a commercial nonlinear solver (MSC Marc). Here, the validity of the proposed method is demonstrated using a convergence test in the model wing and a cantilever bending test.

#### 3.5.1. Convergence of the Model Wing

Figure 6a,b shows the proposed model wing and the leading edge only, respectively. Note that the leading edge in Figure 6a is the same as that in Figure 6b. As shown, the left ends of these models are fixed and a point force 0.01 N in the out-of-plane direction is applied to the right ends of their leading edges. The value of the point force is determined such that the bending displacement of the right end of the leading edge is approximately 10% of its length, which is the large bending case. Table 1 summarizes the geometrical and material properties. The proposed solution procedure is applied to the model wing with the various Young’s moduli of the wing membrane *E*_m_.

As shown in Figure 7, the proposed model wing in Figure 6a (or the monolithic beam-shell structure) converges to the leading edge only in Figure 6b (or the conventional beam structure) continuously as *E*_m_ decreases or when the VMI disappears. That is, the present solution of the monolithic beam–shell structure is consistent with the well-known solution of the conventional beam structure. This result verifies that the proposed method was adequately implemented.

#### 3.5.2. Bending of a Cantilever

Figure 8 shows the monolithic beam–shell structure for the cantilever, where the same cantilever is modeled using beam and shell finite elements (the beam–shell hybrid cantilever). The cantilever is fixed at the left end and the point force in the out-of-plane direction is applied to the right end. Table 2 summarizes the geometrical and material properties of the beam–shell hybrid cantilever. As shown, both the geometrical and material properties for the beam and shell parts are the same as each other. We use the proposed solution procedure to solve this structure. In the mesh, the numbers of elements are 80 for the shell and 20 for the beam.

This structure can be considered as the standard cantilever beam structure or the standard shell structure, of which the geometrical and material properties are summarized in Table 2. Note that the Young’s modulus of the beam structure or the shell structure is twice that of the beam and shell parts in the beam–shell hybrid cantilever. We use the analytical solution based on the Euler–Bernoulli beam theory and the nonlinear finite element procedure for the shell structure.

Figure 9 shows the bending displacement at the cantilever free end. As shown, for the smaller load, the present solution for the monolithic beam–shell structure is indistinguishable from the theoretical beam solution and the conventional numerical solution from the finite element shell analysis. On the contrary, as the load increases, the present solution for the monolithic beam–shell structure differs from the theoretical beam solution because of the geometrical nonlinearity, while it is indistinguishable from the conventional numerical solution from the finite element shell analysis since both these numerical solutions consider the geometrical nonlinearity. This result validates the accuracy of the proposed method.

## 4. Setup of Flapping Insect Wing Model

### 4.1. Shell Structure Representing Wing Membrane

The wing length *R*_w_ and wing chord length *C*_w_ are given as 11.3 mm and 3.11 mm, respectively, from the average values from actual hoverflies [56]. The wing area *S*_w_ is given by their multiplication. Furthermore, the wing weight *m*_w_ is given as 0.350–0.543 mg from the values from actual hoverflies [44,56], and the wing mass density *ρ*_w_ is given as 1.2 g/cm^3^ from actual data for an insect cuticle [57]. The wing volume *V*_w_ can be evaluated as *m*_w_/*ρ*_w_, and the total volume of veins *V*_v_ can be evaluated as the sum of the volumes of the significant veins. The diameters and lengths of the significant veins [58] were used to calculate these volumes. Therefore, the total volume of the wing membrane *V*_m_ is given as *V*_m_ = *V*_w_ − *V*_v_, and the mean thickness of the wing membrane *h*_m_ is given as *h*_m_ = *V*_m_/*S*_w_. It follows from this evaluation that *h*_m_ is estimated as 1.4–5.5 μm. The present evaluation can be validated as follows: We also evaluated a chord section of freshly killed *Calliphoridae* wings as 4μm measured from a microscope image. In ref. [59], the average membrane thickness from different three wing zones measured from SEM micrographs of cross sections ranged from 1.98 ± 0.76 to 4.74 ± 0.42 for *Schistocerca gregaria* hind wings. Also, the SEM micrographs revealed that the mean thickness of the wing membrane of Acheta domesticus was 6.94 ± 0.23 μm [60]. The wing’s membrane thickness of Nephrotoma appendiculata ranged from 1 to 5μm in ref. [61]. We use *h*_m_ = 2.0 μm as the base case. Young’s modulus and Poisson’s ratio for the wing membrane are given as *E*_m_ = 1.0 GPa and *ν*_m_ = 0.49, respectively, from actual data for an insect cuticle [57].

### 4.2. Beams Representing the Supports from the Veins

The macroscopic stiffness of insect wings can be evaluated from bending and torsion tests on actual insects, where the wing base is fixed. In previous studies, the bending and torsional stiffnesses along the wing length are given as *G*_s_ = 3.5 μN m^2^ [27] and *G_θ_* = 0.013 μN m^2^ [14], respectively, while the bending stiffness along the wing chord is given as *G*_c_ = 0.20 μN m^2^. In actual insects, the supporting area in the leading edge dominates the stiffness along the wing length, while the supporting area in the trailing edge dominates the stiffness along the wing chord [28]. Therefore, the bending and torsional stiffnesses of the beam representing the support from the veins in the leading edge (leading-edge beam) are given as *G*_s_ and *G_θ_*, respectively, while the bending stiffness of the beam representing the support from the veins in the trailing edge (trailing-edge beam) is given as *G*_c_. The diameter of the section of the beams representing the intermediate and supplementary supports is given as *d*_v_ = 48 μm, which is based on the most significant vein in these domains [58]. The Young’s modulus and Poisson’s ratio are given as *E*_v_ = 1.0 GPa and *ν*_v_ = 0.49, respectively, from the material properties of an insect cuticle [57].

### 4.3. Boundary Conditions

The node at the base of the wing connects three beams (the left and upper corner points in Figure 1). Therefore, a fixed boundary condition for all degrees of freedom is imposed at this node, as shown in Figure 1. This is because the bending and torsion tests that gave the same *G*_s_, *G_θ_*, and *G*_c_ fixed the base of actual insect wings [14,27]. The aerodynamic pressure acting on the wing surface is given using Equation (1). The parameters in these equations are given as *C*_D_ = 1.19, which correspond to that for a flat plate with an aspect ratio of 4, *ρ*^f^ = 1.2 × 10^−3^ g/cm^3^, *Φ* = 108°, *T_φ_* = 1/*f_φ_*, and *f_φ_* = 161 Hz [9].

## 5. Numerical Analysis of Flapping Insect Wing Model

### 5.1. Basic Validity of the Proposed Model

Figure 10 shows the deformation of the model wing given by the finite element analysis using the setup described in the previous section. As shown in this figure, a positive camber appears over a large part of the model wing. Figure 11 shows the z-displacement *u_z_* of the wing chord and the wing’s sectional shape at the positions of 50%, 70%, and 90% of the wing length as measured from the base. As shown in this figure, the magnitude of the deformation gradient ∂*u_z_*/∂*y* near the leading edge increases as this position moves away from the base. This is because the moment given by the pressure causes an anticlockwise torsion in the wing about the leading edge.

The black curve in Figure 12 shows the distribution of the camber along the wing length. This result shows good agreement with that seen in observation of actual hoverflies [49]. Furthermore, the feathering angle *θ* of 27° at the wingtip is close to that seen in actual observations. As described in the previous section, the estimated thickness of the wing membrane *h*_m_ ranges from 1.4 to 5.5 μm. Therefore, the effect of the thickness on the camber is evaluated in this range.

Figure 12 shows the camber in the cases of *h*_m_ = 3.0–5.0 μm as well as the base case of *h*_m_ = 2.0 μm. As shown in this figure, the camber distribution in the cases of *h*_m_ = 3.0–5.0 μm is also close to that in the actual observations. The similarity in the magnitudes of feathering and cambering demonstrates the validity of the proposed modeling method. Furthermore, as the thickness of the wing membrane increases, the relative effect of the wing veins compared with the wing membrane influence decreases. As shown in Figure 12, the camber decreases as *h*_m_ increases. This result demonstrates the necessity of significant veins in the wing’s membrane or the VMI to create the wing camber. It follows from these demonstrations that the VMI will be a mechanical basis of the cambering of flapping insect wings.

The most recent literature for the structural properties of a wing’s material [62] has shown that the Young’s modulus of the vein is normally higher than that of the membrane, and the Young’s modulus of the vein is nearly 5 GPa when fresh and 9 GPa when dry. In this study, the modulus of the membrane and the veins is set to be 1 GPa. Figure 12 also shows that the magnitude of the camber increases monotonically as the stiffness ratio of the vein to the membrane increases. Hence, if the Young’s modulus of the veins is set to 5 Gpa or much higher than the 1 GPa of the membrane, the magnitude of the camber is expected to be increased.

Details of the mechanical roles of the veins will be discussed in the following section using a parametric study under the present setup with a base case of *h*_m_ = 2.0 μm.

### 5.2. Effect of Central and Tip Veins

To discuss the roles of the veins in the wing’s central and tip domains, two cases are considered. As shown in Figure 13, in Case A, the beam representing the wingtip veins is removed, while in Case B, the beam representing the central wing veins is removed.

The deformation and camber distribution of the model wing for each case are shown in Figure 14 and Figure 15, respectively. As shown in these figures, the camber in Case A increases in the range from the wing center to the tip compared with the base case (Figure 1). This is because the beam representing the wingtip domain increases the stiffness to restrict the deformation in the vicinity of the area from the wing center to the tip.

In contrast, as shown in these figures, the camber in Case B is reduced significantly. Figure 16 shows the distribution of the difference of the *z*-displacement between the base case and Case B. As shown in this figure, the beam representing the wing’s central veins has two major effects: One is to increase the *z*-displacement at the middle of the chord along the length. The other is a decrease in the *z*-displacement in the vicinity of the trailing edge, especially near the right corner.

These effects cause the beam representing the central veins to increase the camber. The reason this beam increases the *z*-displacement at the middle of the chord along the length is probably that this beam transmits the pressure moment from the middle of the chord to the tip. Conversely, the reason this beam decreases the *z*-displacement in the vicinity of the trailing edge is probably that this beam increases the stiffness in the vicinity of the trailing edge.

### 5.3. Effect of Torsional Flexibility of Veins near the Leading Edge

To discuss how the torsional flexibility of the veins in the vicinity of the leading edge affects the cambering of the wing, the torsional stiffness of the beam at the leading edge was changed from the base case (*G_θ_* = 0.013 μN m^2^) to a case using a torsional stiffness 10 times smaller than the base case and a case using a torsional stiffness 10 times larger than the base case. Figure 17 shows the sectional shapes of the deformed wing at the position 50% of the wing length, and Figure 18 shows the distributions of the camber along the wing length. As shown in these figures, the camber increases as the torsional rigidity of the beam at the leading edge decreases. The reason for this is explained below.

As shown in Figure 17, the initial bending angle of each chord at the leading edge increases as the torsional stiffness of the beam at the leading edge decreases. This is because the torsional deformation of the leading edge increases as the torsional stiffness of the beam at the leading edge decreases. Therefore, the bending displacement of each chord increases as the torsional stiffness of the beam at the leading edge decreases. Simultaneously, the beam representing the central veins reduces the displacement of each chord in the vicinity of the trailing edge, as discussed in the previous section. It follows from these effects that the camber increases as the torsional stiffness of the leading edge decreases.

Figure 19 shows the relationship between the mean camber and the second polar moment of inertia of the area for the beam at the leading edge. As shown in this figure, the mean camber decreases monotonically as the second polar moment of the area increases. Note that the mean camber converges to specific values for very small and large second polar moments. This is because the support of the beam at the leading edge for each wing chord converges to a fixed support for a very large second polar moment, while it converges to a pin support for a very small second polar moment.

### 5.4. Effect of the Range of the Central Veins

The central veins play an essential role in cambering, as discussed in the previous section. The range of the central domain supported by these veins can be considered as their characteristic parameter. Therefore, two cases based on Case A in the previous section are considered to discuss the effect of the range of the central veins on cambering, as shown in Figure 20.

In one case, the upper end of the beam representing the central veins (central beam) is moved (meshes A, B, C, and D), and the displacement of this end is up to about 50% of the length of the leading edge. Figure 21 shows the deformation of the model wing in mesh D, where the upper end of the central beam exhibits the maximum displacement. This deformation is similar to that of the original mesh (Figure 14a). Figure 22 shows the camber distributions along the wing length for meshes A, B, C, and D, as well as the original mesh. As can be seen, this type of change in the range of the central beam does not have much effect on the magnitude of the camber. This is because the camber on the right-hand side of the model wing imposes the camber on the left-hand side of the wing.

In contrast, in the other case, the lower end of the beam is moved (meshes E, F, G, and H) and a significant decrease in the camber occurs if the displacement of this end exceeds a specific limit. As shown in Figure 23a,b, the deformation of the model wing for meshes E, F, and G is similar to that of the original mesh (Figure 14a), while the deformation of the model wing for mesh H is quite different, as shown in Figure 23c, where the mode of the deformation is shifted. It follows from this shift that the camber is significantly changed from mesh G to mesh H, as shown in Figure 24. Regardless of the absence of support from the beam, the shell in the vicinity of the right lower corner in meshes E, F, and G does not show a bending deformation like that in Figure 23b. This is because the camber increases the bending rigidity around this domain.

## 6. Concluding Remarks

We presented the VMI concept to clarify the mechanical process of how elastic deformation of flapping insect wings caused by aerodynamic pressure results in their significant cambering. We proposed a numerical method for the VMI that consists of a shape simplification model wing and a nonlinear monolithic solution procedure. We incorporated data from actual insects into the model wing and demonstrated significant cambering as well as feathering, of which magnitudes were equivalent to those of the actual insects, and the importance of the VMI in their generation. Furthermore, several points concerning the mechanical roles of the veins in cambering were revealed:(a)The veins crossing the central domain diagonally transmit the bending moment due to the aerodynamic pressure, such that the bending deformation increases at the middle of the chord along the length. Simultaneously, the central veins increase the bending stiffness of the wing in the vicinity of the trailing edge to restrict the bending deformation in this area. Both of these effects increase the camber effectively.(b)The veins near the wingtip increase the bending stiffness to restrict the camber in this domain.(c)The veins near the leading edge increase the initial bending angle of the wing chord in this area as their torsional flexibility increases. This effect increases the camber under the restriction of the bending deformation in the vicinity of the trailing edge given by the veins in the central domain.(d)The minimum function of the central veins required for cambering the whole wing is that they can form a local camber at the wingtip. In turn, this local camber imposes a deformation on the rest of the wing such that it causes a global camber over the whole wing.

This mechanical understanding of the roles of veins in cambering will provide a design basis for micro and nano aerial vehicles that mimic the flapping flight of insects. It is possible to verify the results via a micro-wing design implementation with real flight experiments [63,64,65], which will be our future work. Insect-mimetic micro wings have been fabricated in Refs. [58,61], but at this time it seems that there is no insect-scale artificial wing that can present a positive camber comparable with that of actual insects, to our best knowledge. The proposed model has a 2.5-dimensional structure, which is suitable for polymer micromachining based on the photolithography. Hence, the proposed model is promising for fabricating insect-mimetic micro wings with the sufficient positive camber. Torsional flexibility affects the aerodynamic performance of insect wings, and the twisting influences the distribution of lift and drag along the wingspan, contributing to overall flight efficiency. Hence, the correlation of the twisting due to the torsional flexibility with different flight phases will be a basis for understanding take-off, landing, hovering, evasive maneuvers, etc. Direct numerical modeling will be a promising approach for exploring this issue, while it includes challenging problems of computational mechanics. For this purpose, the proposed model wing will be incorporated into state-of-the-art numerical methods for insect flapping flight, such as the strongly coupled FSI analysis method [10,66], the maneuvering analysis method [67], and the coupling method of global and local meshes [68] in our future work.

## Figures and Tables

**Figure 1 biomimetics-08-00571-f001:**
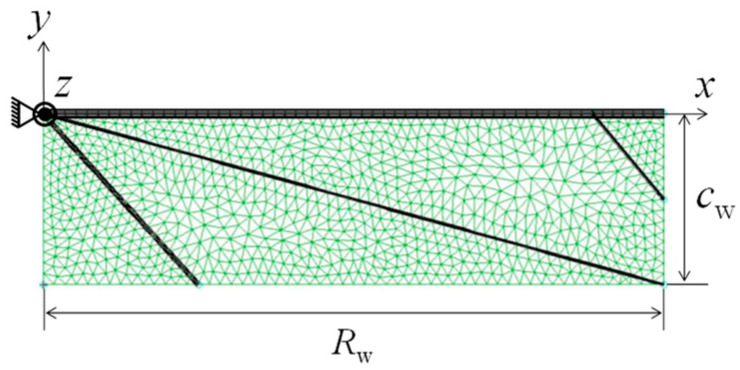
Shape simplification modeling of an insect wing. *R*_w_ and *C*_w_ express the wing and chord lengths, respectively. The wing membrane is discretized using triangular shell elements, which are colored green, while the beams corresponding to the supports from the veins are indicated by bold black lines.

**Figure 2 biomimetics-08-00571-f002:**
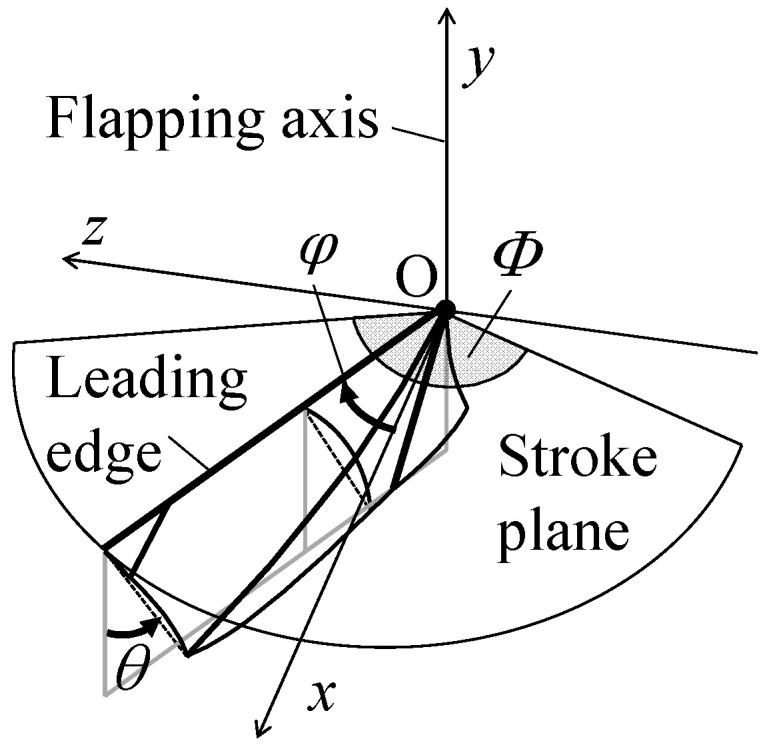
Schematic of model of a flapping insect wing.

**Figure 3 biomimetics-08-00571-f003:**
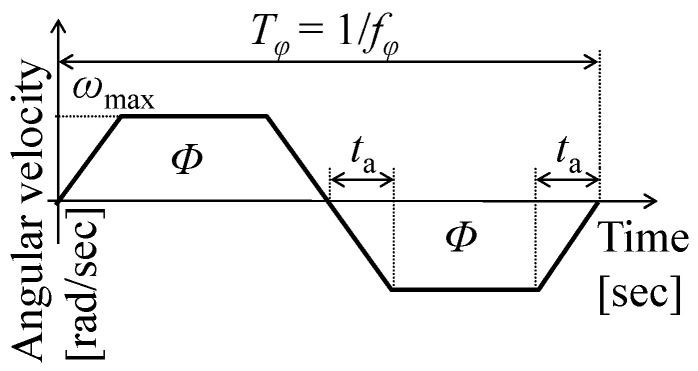
Time history of flapping angular velocity. *Φ*, *t*_a_, and *ω*_max_ express the stroke angle, the acceleration time, and the maximum flapping angular velocity, respectively.

**Figure 4 biomimetics-08-00571-f004:**
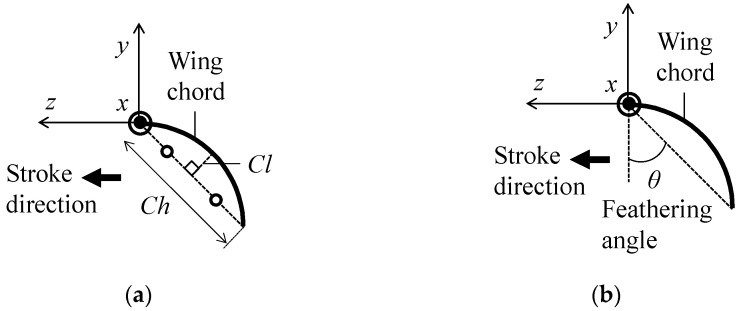
Definitions of (**a**) camber and (**b**) torsion. The bold lines indicate the wing chord. The bold arrows indicate the flapping direction.

**Figure 5 biomimetics-08-00571-f005:**
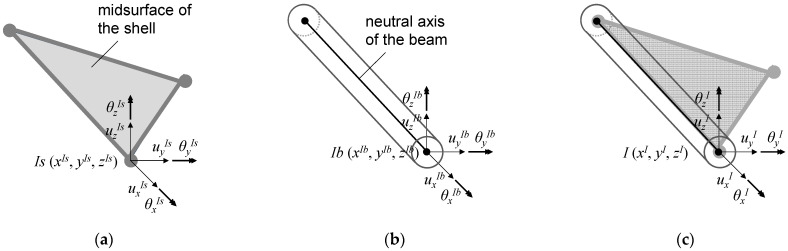
(**a**) Shell element, (**b**) beam element, and (**c**) the same arrangement of their nodes.

**Figure 6 biomimetics-08-00571-f006:**
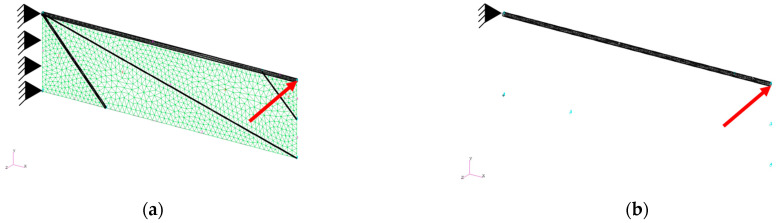
(**a**) Shell and beam finite elements with 906 nodes for the proposed model wing and (**b**) beam elements with 58 nodes for the leading edge. The red arrow indicates the point force.

**Figure 7 biomimetics-08-00571-f007:**
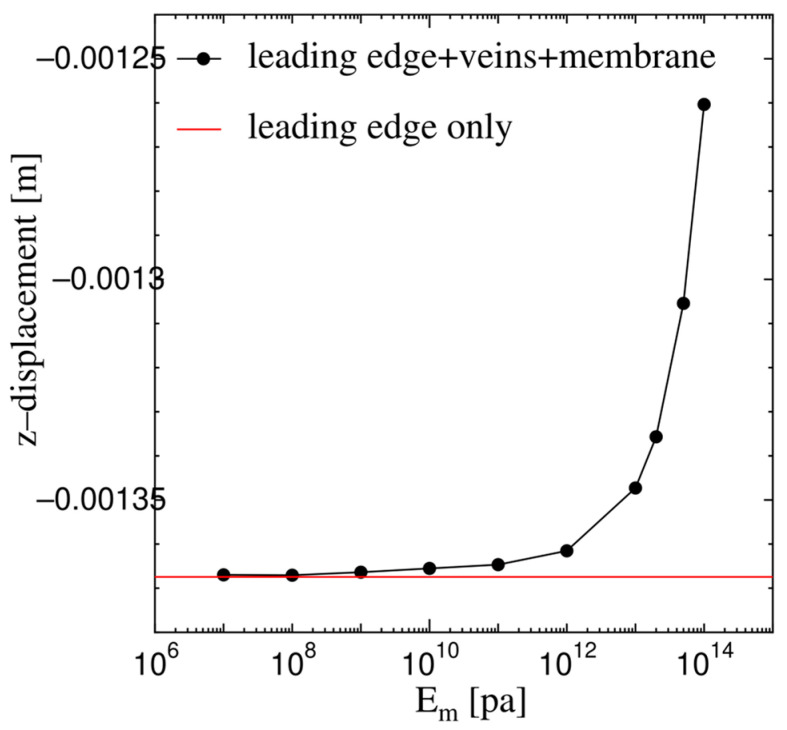
Convergence of the proposed model wing to the leading edge only.

**Figure 8 biomimetics-08-00571-f008:**
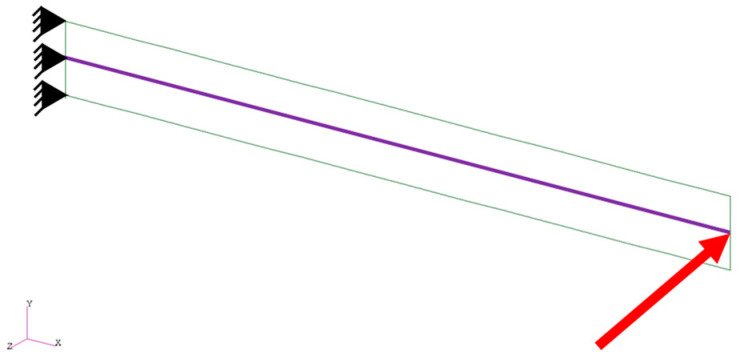
The beam–shell hybrid structure for the cantilever. The red arrow indicates the point force.

**Figure 9 biomimetics-08-00571-f009:**
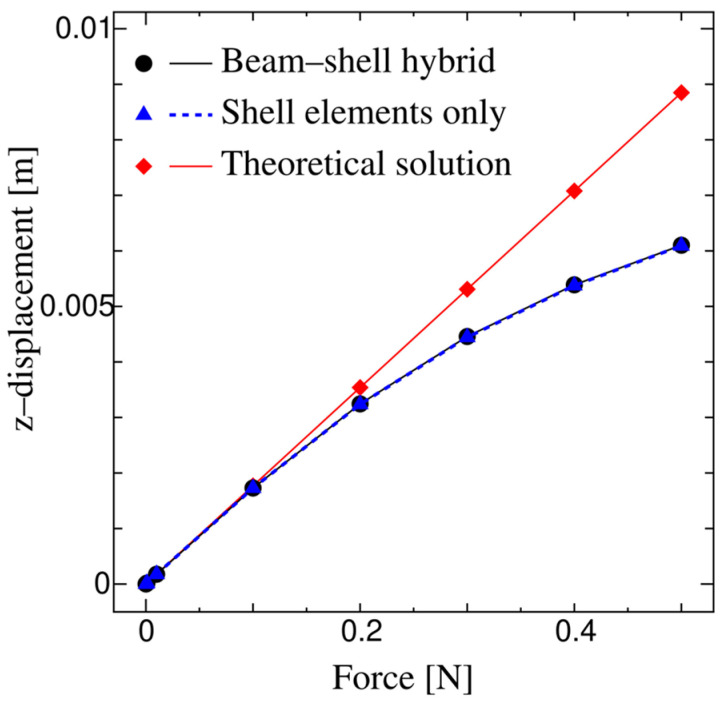
Bending displacements at the free end of the cantilever for the beam–shell hybrid structure, the beam structure, and the shell structure.

**Figure 10 biomimetics-08-00571-f010:**
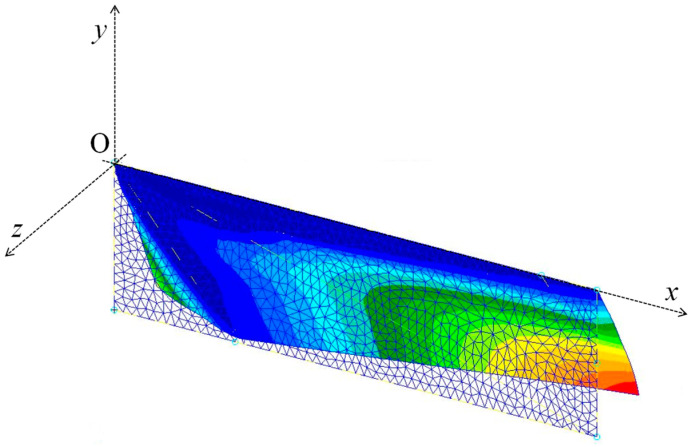
Deformation of model wing in the base case. The color contours show the magnitude of displacement in the *z*-direction.

**Figure 11 biomimetics-08-00571-f011:**
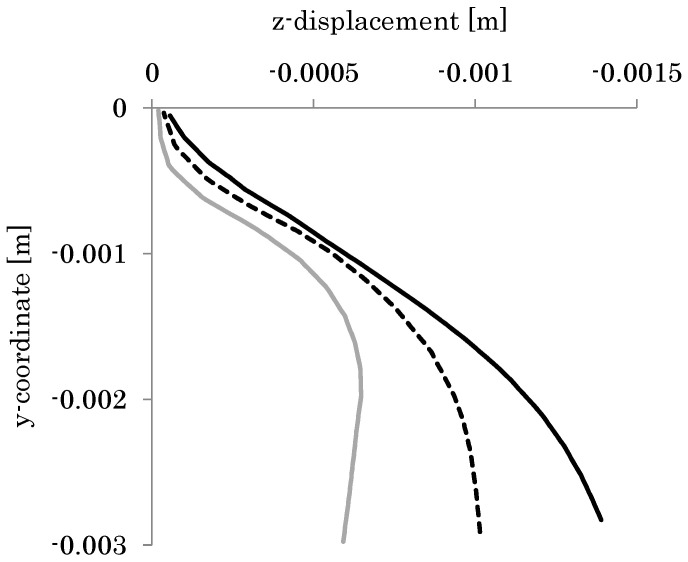
Wing chord displacements in the *z*-direction at the positions of 50% (gray), 70% (dotted), and 90% (black) of the wing length from the wing base.

**Figure 12 biomimetics-08-00571-f012:**
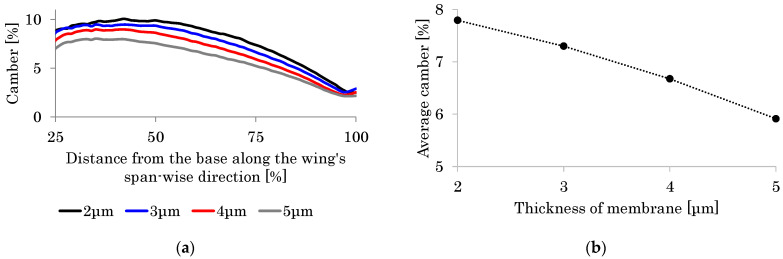
Cambers in the base case (membrane thickness *h*_m_ = 2 μm, black curve) and the other membrane thickness. (**a**) The horizontal axis shows the distance from the wing base along the wing length divided by the total wing length, and the vertical axis shows the camber (*h*_m_ = 5 μm, gray curve; *h*_m_ = 4 μm, red curve; *h*_m_ = 3 μm, blue curve). (**b**) The horizontal axis shows the membrane thickness *h*_m_ and the average camber.

**Figure 13 biomimetics-08-00571-f013:**
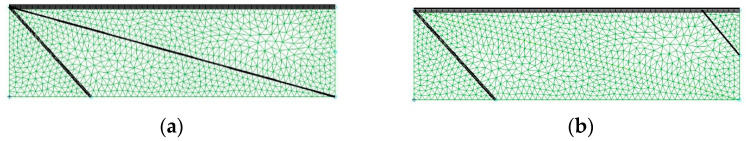
Meshes for (**a**) Case A and (**b**) Case B. The number of nodes is 913.

**Figure 14 biomimetics-08-00571-f014:**
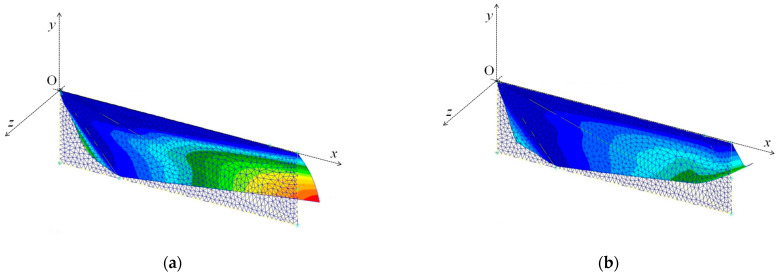
Model wing deformations in (**a**) Case A and (**b**) Case B. The color contours show the magnitude of the displacement in the *z*-direction.

**Figure 15 biomimetics-08-00571-f015:**
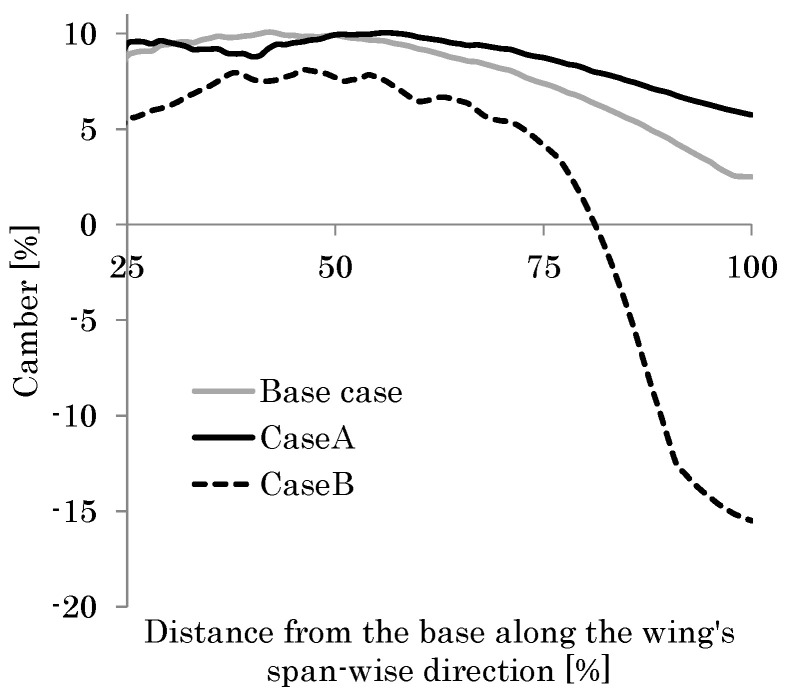
Camber distribution along the wing length. The gray curve indicates the base case (Figure 1), the black curve indicates Case A, and the dotted curve indicates Case B.

**Figure 16 biomimetics-08-00571-f016:**
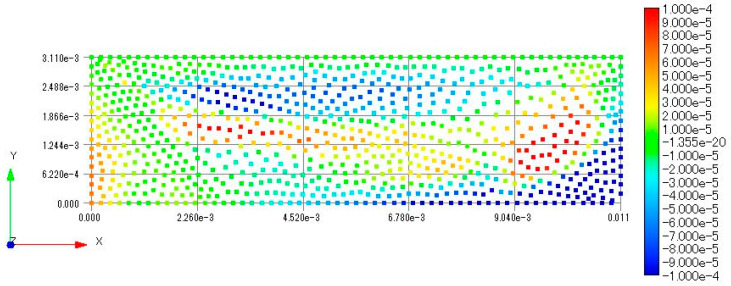
Distribution of the difference of the z-displacement between the base case and Case B.

**Figure 17 biomimetics-08-00571-f017:**
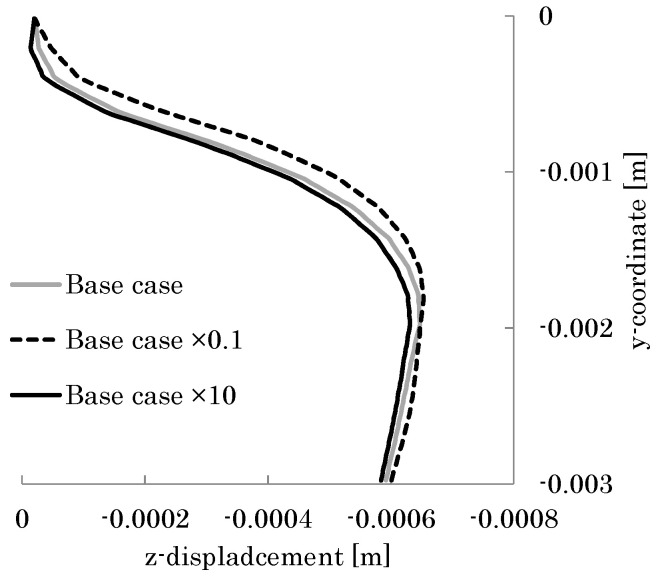
Displacements in the *z*-direction of the wing chord at the position of 50% of the wing length. The gray curve indicates the base case. The dotted curve indicates the case using the torsional stiffness 10 times smaller than the base case. The black curve indicates the case using a torsional stiffness 10 times larger than the base case.

**Figure 18 biomimetics-08-00571-f018:**
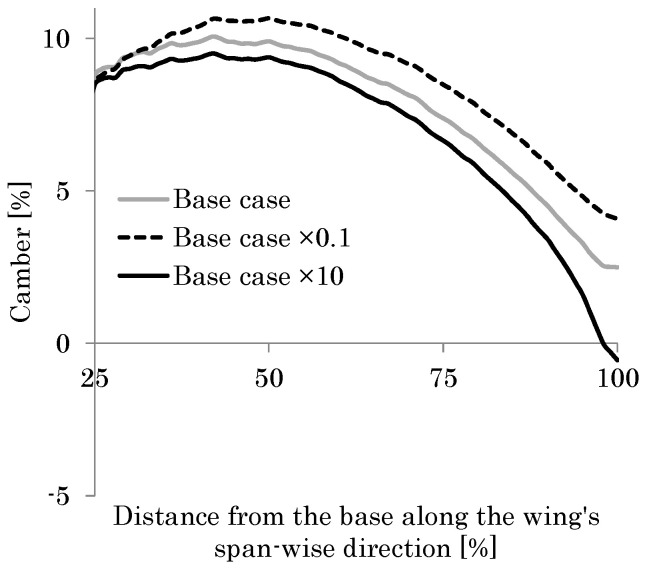
Camber distribution along the wing length. The dotted curve indicates the case using a torsional stiffness 10 times smaller than the base case, which is indicated by the gray curve. The black curve indicates the case using a torsional stiffness 10 times larger than the base case.

**Figure 19 biomimetics-08-00571-f019:**
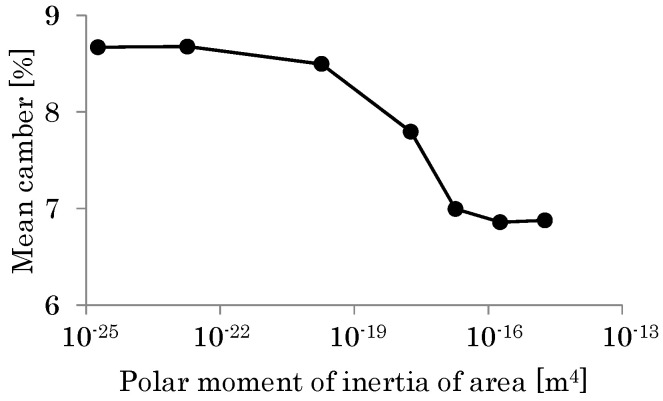
Relationship between the mean camber and the polar moment of inertia of the area of the beam at the leading edge.

**Figure 20 biomimetics-08-00571-f020:**
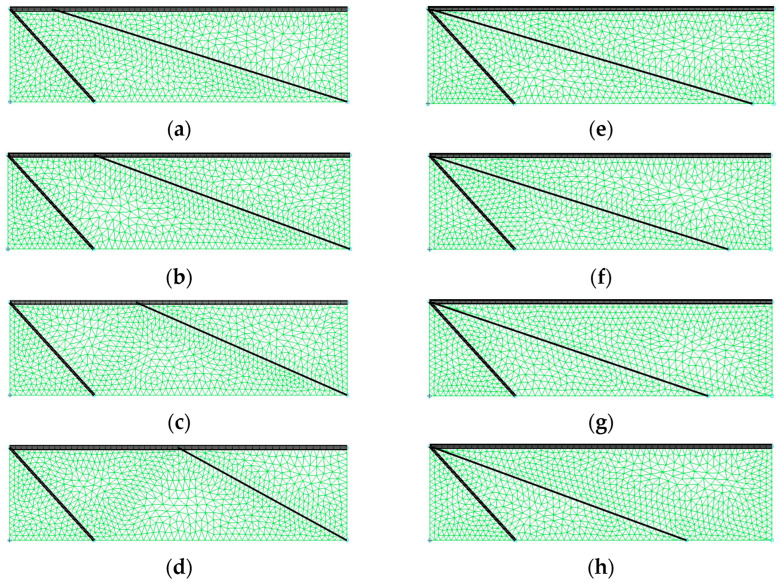
Meshes used to consider the effect of the range of the beam representing the central veins. In (**a**) mesh A, (**b**) mesh B, (**c**) mesh C, and (**d**) mesh D, the upper end of the beam is moved along the leading edge from left to right to decrease the range of the beam along the wing length. In (**e**) mesh E, (**f**) mesh F, (**g**) mesh G, and (**h**) mesh H, the lower end of the beam is moved along the trailing edge from right to left to decrease the range of the beam along the wing length.

**Figure 21 biomimetics-08-00571-f021:**
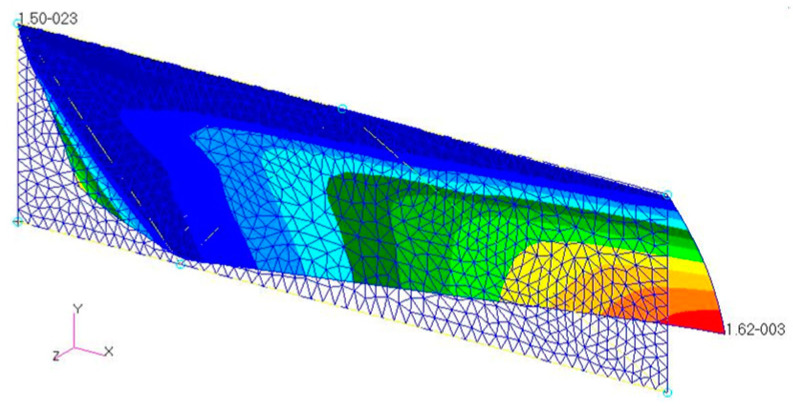
Deformation of the model wing in the case of mesh D. The color contours show the magnitude of the displacement in the *z*-direction.

**Figure 22 biomimetics-08-00571-f022:**
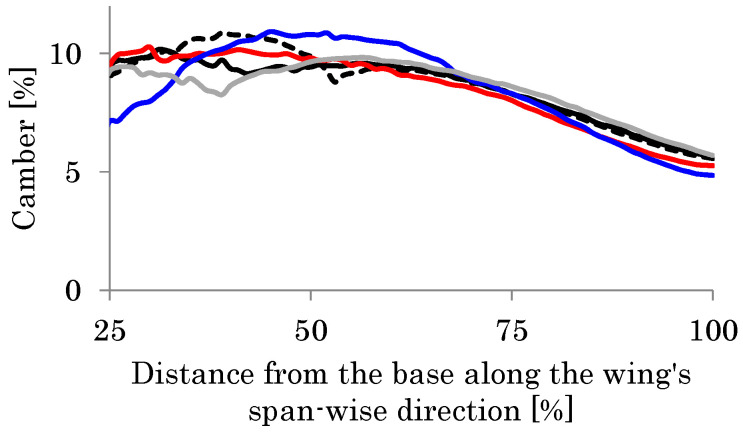
Camber distribution along the wing length. The gray curve indicates the original mesh (Figure 1), the black curve indicates mesh A, the dotted curve indicates mesh B, the red curve indicates mesh C, and the blue curve indicates mesh D.

**Figure 23 biomimetics-08-00571-f023:**
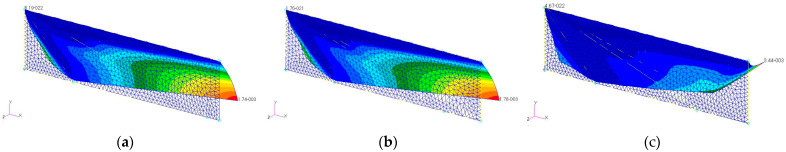
Deformation of the model wing in the cases of (**a**) mesh E, (**b**) mesh G, and (**c**) mesh H. The color contours show the magnitude of the displacement in the *z*-direction.

**Figure 24 biomimetics-08-00571-f024:**
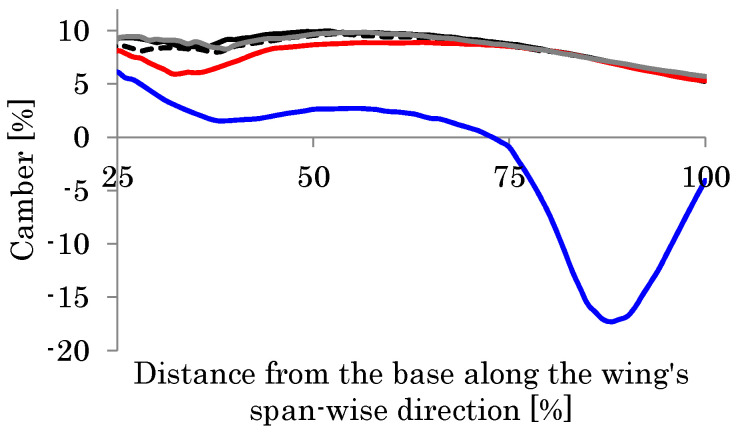
Camber distribution along the wing length. The gray curve indicates the original mesh (Figure 1), the black curve indicates mesh E, the dotted curve indicates mesh F, the red curve indicates mesh G, and the blue curve indicates mesh H.

**Table 1 biomimetics-08-00571-t001:** Geometrical and material properties of the model wing in the convergence test.

Properties	Values
Wing length [m]	0.0113
Wing chord length [m]	0.00311
Young’s modulus of leading edge [GPa]	100
Diameter of the section of leading edge [μm]	163
Young’s modulus of root vein [GPa]	60.0
Diameter of the section of leading edge [μm]	90.0
Young’s modulus of center and support vein [GPa]	1.00
Diameter of the section of leading edge [μm]	48.0
Thickness of wing membrane [μm]	2.00
Poisson’s ratio	0.490

**Table 2 biomimetics-08-00571-t002:** Geometrical and material properties of three structures of the cantilever.

Properties	Beam-Shell Hybrid	Beam Only	Shell Only
Length [m]	0.0113		
Width [m]	0.00113		
Thickness [m]	0.000113		
Young’s modulus of beam [GPa]	100	200	NA
Young’s modulus of shell [GPa]	100	NA	200
Poisson’s ratio	0.0		

## Data Availability

The datasets analyzed during the current study are available from the corresponding author upon reasonable request.

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
