# Peer review of "Vein–Membrane Interaction in Cambering of Flapping Insect Wings"

_biomimetics, 2023, doi:10.3390/biomimetics8080571_

Round 1

Reviewer 1 Report (Previous Reviewer 4)

Comments and Suggestions for Authors

The manuscript can be accepted for publication in current form.

Author Response

Many thanks for taking your time to review our manuscript. We wish to express our appreciation to you for giving us the recommendation for publishing our manuscript in its present form.

Reviewer 2 Report (New Reviewer)

Comments and Suggestions for Authors

Dear authors,

the article is part of a study on adaptive systems, in particular presents aerodynamic analyzes on the wing structures of insects. The topic is certainly interesting and can arouse curiosity.

Below some aspects that in my opinion need to be improved:

- insert a list of abbreviations and symbols used;

- represent the trend of stiffness (flexural EI and torsional GJ) as a function of wing span;

- is it possible to represent the elastic axis of the beam model?

- describe the programs and tools used for FEM analysis (including the solver, linear, nonlinear?). From the pictures it looks like Patran/Nastran;

- what are the limitations of classical theories when analyzing such a small wing structure? Please substantiate with strong references too;

- is there the possibility of experimentally validating these calculations?

- to better indicate the innovation introduced with this research towards the community, compared to what is already present in the literature.

My regards.

Comments on the Quality of English Language

A general revision could be good.

Author Response

Reviewer 3 Report (New Reviewer)

Comments and Suggestions for Authors

This study proposes a VMI concept to clarify the mechanical process of how elastic deformation of flapping insect wings caused by the aerodynamic pressure results in their cambering. Although the obtained results are intriguing, some revisions are necessary before considering this paper for publication. The following suggestions are provided:

1. The main contributions of this paper should be clearly presented in the Introduction section.

2. The article lacks a clear demonstration of the main process involved in the proposed method. It is important to establish a clear connection between the innovative points presented in the paper.

3. In Section 4, necessary demonstrations should be plotted to show shell structure and boundary conditions.

4. In Fig. 11, the corresponding meanings of these lines should be given as labels in the figure.

5. Some references are relatively out of date, papers in recent three years should be cited.

Comments on the Quality of English Language

Moderate editing of English language required.

Round 2

Reviewer 2 Report (New Reviewer)

Comments and Suggestions for Authors

Dear authors,

the paper sounds really good now. By my side, it is ready for acceptance.

My kindest ragards and good luck for the future experimental works.

Comments on the Quality of English Language

No particular issues. A general revision by the editor office is enough.

Reviewer 3 Report (New Reviewer)

Comments and Suggestions for Authors

No more comments.

This manuscript is a resubmission of an earlier submission. The following is a list of the peer review reports and author responses from that submission.

Round 1

Reviewer 1 Report

Comments and Suggestions for Authors

At first sight I, as a biologist, considered this paper to be typical of a non-biological approach in which a crude model of a complex reality is proposed and analysed with the implication that the model is a good representation of reality, but is never validated by comparison with the biological reality. This would then turn out to be irrelevant to biology. But let that rest for a while.

The model is undoubtedly simple. But why is it 'derived' from one of the most complex wings amongst the insects? Coombes [27] gives many other basal morphologies that would be far simpler as models. Some justification is needed.

The proposed model is then derived, and bears little resemblance to the biological type. Long experience tells me that the wobbly trajectories of the veins in the fly wing are not there for show. They have an important functional origin. No reason is given why straight veins are valid. We need some justification.

Analysis of this crude model proceeds. I have no argument with this. Indeed I have to take it on faith that it's OK. For this reason all the maths should be sequestered into a different part of the paper, either as a separate follow-up section, or into text boxes in the appropriate parts of the paper. Whatever - the maths should be separated out and only the results revealed in the general text, expressed non-mathematically or graphically. I need some sort of flag to tell me when it's safe to continue reading the main argument!

At this point the argument should revert to the starting point. Has the original stated goal (camber) been reached. Indeed, since the hoverfly wing, like many insect wings, generates its lift by means to a leading-edge vortex, is the camber of any significance at all.

At this point a reference [54] is brought in which should have been one of the first to be revealed. Not only does it do the mechanical tests that might validate the model, at least in terms of how close a model (of dubious validity) can come to the original wing it claims to copy, but its performance is comparable. I quote from [54]:

The maximum lift achieved with the [carbon fibre] wing, 6.9 x 102 mN, was greater than that of the polymer wing, 5.9 x 102 mN. These results suggest that hoverflies can utilize passively pronation and supination during hovering flight, and that for similar wing flapping motions and wing planforms, a rigid wing can produce larger lift. The effect of wing flexibility on drag and efficiency, however, require further investigation.

Unfortunately it also throws the usefulness of the model into doubt since the better-performing model is rigid and so does not develop camber! Also the venation of the model in [54] is very similar to that used in this paper. Coincidence or what?

Overall I am not convinced by this paper. The presentation does not help the reader to consider that a valid model is either derived or analysed, and at the end one is faced with a possible trade-off between a more flexible wing that may be lighter but performs less well and produces an apparently irrelevant camber.

Comments on the Quality of English Language

In English, most adjectives and adverbs are singular irrespective of whether the accompanying noun is singular or plural. There is a proliferation of spurious 'plural' adjectives in this paper that should be corrected. For instance, in line 60 "varieties" should be changed to "variety". "Varieties" would refer to different genotypes of a species. Also the typography is faulty in places, notably in line 43 where the plural 's' of "wing's" is pushed into the next line!

Otherwise not too bad.

Reviewer 2 Report

Comments and Suggestions for Authors

The manuscript represents a study of the author's team on vein-membrane interaction in the cambering of flapping insect wings. They investigate the vein membrane interaction process. For this purpose, the model of the flapping insect wing was created and evaluated in the simulation environment. After successful verification, the author's team conducted simulations, which results are discussed in a separate chapter of the manuscript. The authors are providing a design basis for micro and nano-aerial vehicles that mimic the flapping flight of insects.

The introduction is fine, and relevant sources are cited. The separate chapters explain the creation of the model and mathematical backup. The results chapter analyses simulation outcomes.

The article is well-written, technically sound and readable with ordinary effort. Its length is adequate, and it is well organised.

As I found no issues, I recommend publishing it in its present form.

Author Response

Many thanks for taking your time to review our manuscript. We wish to express our appreciation to you for giving us the recommendation for publishing our manuscript in its present form.

Reviewer 3 Report

Comments and Suggestions for Authors

This paper describes the vein-membrane interaction (VMI) concept to clarify the mechanical process of elastic deformation of flapping insect wings caused by the aerodynamic pressure that results in their significant cambering. A numerical method for the VMI is proposed which consists of a model wing and a nonlinear monolithic solution procedure. The data is incorporated from actual insects into the model wing, and demonstrated significant cambering as well as feathering, whose magnitudes were found to be equivalent to those of the insects analyzed during the study.

In section 2.1, a shape simplification model is used for the wing approximation. How much inaccuracies are induced via this simplification?

In Eq. (1a), what value of drag coefficient is used? What fly regimes are considered?

In the nonlinear analysis framework (3.3), how many degrees of freedom are considered? Please explain.

Torsional flexibility affects the aerodynamic performance of insect wings. The twisting of veins influences the distribution of lift and drag along the wingspan, contributing to overall flight efficiency. How can it be correlated with different flight phases e.g. take-off, landing, hovering, Evasive Maneuvers etc?

A variety of mini, micro and nano aerial vehicles are in use today. Is it possible to verify the results via a micro-wing design implementation with real flight experiment?

Please add more recent works as few references are added from the last 5 years work.

The paper language requires attention. It must be revised by a technical English expert. Abstract and Conclusion need considerable revision.

Comments on the Quality of English Language

English language must be improved in the revised version.

Reviewer 4 Report

Comments and Suggestions for Authors

Comments on the Quality of English Language

Language needs minor editing.

Round 2

Reviewer 1 Report

Comments and Suggestions for Authors

Reply to Reviewer 1

Dear Reviewer,

Many thanks for your valuable comments and suggestions that have led to significant improvement in the presentation and quality of this manuscript. In what follows, we shall detail the changes we have made on this manuscript.

Sincerely,

My comments are highlighted in red

The model is undoubtedly simple. But why is it 'derived' from one of the most complex wings amongst the insects? Coombes [27] gives many other basal morphologies that would be far simpler as models. Some justification is needed.

Reply:

We wish to thank the reviewer for this comment. Main reason why we consider the dipteran insect is that it has been typically used in many mechanical and biological studies, leading the availability for the mechanical and morphological data.

There is no such thing as a typical Dipteran wing; the wing of a hoverfly can certainly NOT be considered as typical of the Diptera. More primitive Diptera such as the Nematocera have much simpler wings with relatively straight veins resembling wings of other orders of insect – hence more typical. As for mechanical properties, these are varied later in the paper by a factor of 5 and so cannot be said to be related to measured values from a Syrphid wing.

 For the complexity of flapping insect wings, we used a reduced-order approach, which gives a framework that characterizes the complicated behaviors and a well-defined functional target, and can guide determining the mechanism for the behavior. This point was added to Introduction of the revised manuscript (written using green). Our methodology can be applied for other insect wings which have simpler morphologies if the necessary data is available from previous studies.

There is copious data on locust wings (cf. Wootton at al.) with much simpler venation and much simpler flapping patterns, without the rotation about the leading edge that is so typical of Syrphids. The venation of locust wings is simpler; the mechanical and morphological properties have been more fully and carefully measured. The interest in Syrphid wingss is mainly to account for their sophisticated flight behaviour.

The proposed model is then derived, and bears little resemblance to the biological type. Long experience tells me that the wobbly trajectories of the veins in the fly wing are not there for show. They have an important functional origin. No reason is given why straight veins are valid. We need some justification.

Reply:

We are uncertain as to the meaning of the reviewer’s comment “The wobbly trajectories of the veins have an important functional origin”. If we understand the reviewer correctly in stating a supporting function, then we suggest that the vein’s shape is just one of parameters to determine the stiffness. In the framework of the reduced-order approach, the model wing used the straight beams that have the stiffnesses equal to those of the actual insect wing. The effect of the vein’s shape on the VMI might be interesting, but it is not the issue of this study.

The shape of the veins certainly IS an issue in this study if it is ignored at the same time as claiming that the hoverfly wing is the model. There are many instances in insects and spiders where a non-linear profile has been ignored in order to simplify the analysis, only to find that the non-linearity of the calculated answers is the result of ignoring the precise morphology at the start. The pretence of using the Syrphid wing as a model should be deleted throughout this paper. It has no validity.

Analysis of this crude model proceeds. I have no argument with this. Indeed I have to take it on faith that it's OK. For this reason all the maths should be sequestered into a different part of the paper, either as a separate follow-up section, or into text boxes in the appropriate parts of the paper. Whatever - the maths should be separated out and only the results revealed in the general text, expressed non-mathematically or graphically. I need some sort of flag to tell me when it's safe to continue reading the main argument!

Reply:

We appreciate the reviewer's concerns on this point. However, Section 3 presented the proposed numerical procedure for accurately predicting nonlinear mechanical behaviors of the model wing with the VMI. We believe that if this section is simplified it is difficult for readers to reproduce the contents of this manuscript. Hence, we would like to retain the original text.

Refer to above comment – how do you know that the non-linear behaviour is not due to your ignoring the apparent complexity of the wing’s morphology? The morphological detail gives the desired function. And I didn’t request that the maths should be simplified. For most biologists it will be opaque, requiring more background knowledge and skill than we possess. It serves only to confuse and, perhaps, impress. Present the calculations either as an appendix to the paper, or separately as additional information. The authors gain no advantage by presenting impenetrable modelling.

At this point the argument should revert to the starting point. Has the original stated goal (camber) been reached. Indeed, since the hoverfly wing, like many insect wings, generates its lift by means to a leading-edge vortex, is the camber of any significance at all.

Reply:

The camber’s contribution on the aerodynamic efficiency has been reported by many other studies mainly using CFD and numerical FSI approaches. Please see “The camber increases the aerodynamic performance of insect flapping flight [21–24]” in Introduction.

At this point a reference [54] is brought in which should have been one of the first to be revealed. Not only does it do the mechanical tests that might validate the model, at least in terms of how close a model (of dubious validity) can come to the original wing it claims to copy, but its performance is comparable. I quote from [54]:

The maximum lift achieved with the [carbon fibre] wing, 6.9 x 102 mN, was greater than that of the polymer wing, 5.9 x 102 mN. These results suggest that hoverflies can utilize passively pronation and supination during hovering flight, and that for similar wing flapping motions and wing planforms, a rigid wing can produce larger lift. The effect of wing flexibility on drag and efficiency, however, require further investigation.

Reply:

We are uncertain as to the meaning of the reviewer’s quotation from Ref. [54], since it was the comparison between the realistic polymer wing and the rigid carbon-fiber wing: These wings did not reproduce any positive camber, and the passive pronation and supination or feathering motions were basically given by the torsional spring independent from these wings. Hence, we believe that there is no relevance between this quotation and our study. 

Reference [54} refers to camber three times in the text, related only to the less rigid polymer wing. It was negative, which suggests the model wing was not performing like the biological wing, perhaps because the base of the wing was too stiffly supported.

Unfortunately it also throws the usefulness of the model into doubt since the better-performing model is rigid and so does not develop camber! Also the venation of the model in [54] is very similar to that used in this paper. Coincidence or what?

Reply:

We appreciate the reviewer's concerns on this point.

However, we believe that the first sentence of this comment seems to be based on the different opinion from the previous studies: Many other studies have reported that the camber contributes to the aerodynamic efficiency.

We are uncertain as to the meaning of the second sentence of the reviewer’s comment. If we understand the reviewer correctly in stating the shape similarity with the rigid carbon-fiber (CF) wing in Ref. [54], then we suggest that the rigid CF wing did not consider the VMI proposed in this study or it was a rigid wing and it never reproduced any positive camber, while our model wing considered the VMI or it was a flexible wing and it reproduced the positive camber.

I am not very impressed by the emphasis on the aerolastic origin of camber. Throughout the present paper the assumption is that the veins at the base or root of the wing are firmly joined. This is not the case in an insect wing, where there are three separate hinge points at the root, each with its own musculature capable of changing the hinge characteristics. The wing can therefore have at least three zones over its span, each of which can be moved slightly differently throughout the stroke. Wootton has demonstrated, in simple models, that this can camber the entire wing differently in the up and down strokes. Aerolastic behaviour may not be relevant to the functioning of the wing and its camber, acting only to modulate the behaviour rather than control or generate it passively as this paper maintains. Any model should take this level of control, present in the wings of all insects, into consideration.

In English, most adjectives and adverbs are singular irrespective of whether the accompanying noun is singular or plural. There is a proliferation of spurious 'plural' adjectives in this paper that should be corrected. For instance, in line 60 "varieties" should be changed to "variety". "Varieties" would refer to different genotypes of a species. Also the typography is faulty in places, notably in line 43 where the plural 's' of "wing's" is pushed into the next line! Otherwise not too bad.

Reply:

We wish to thank the reviewer for this comment. We have had the manuscript rewritten by an experienced scientific editor, who has improved the grammar and stylistic expression of the paper. Please also refer the certificate, which is attached as the supplemental material.

Sorry – I wouldn’t employ them!

Comments on the Quality of English Language

A bit better. I agree that for a non-English speaker it can be difficult to replace many words with one - the ideal situation.